# Study on the Preparation, Characterization, and Stability of Freeze-Dried Curcumin-Loaded Cochleates

**DOI:** 10.3390/foods11050710

**Published:** 2022-02-28

**Authors:** Lijuan Chen, Bowen Yue, Zhiming Liu, Yali Luo, Lu Ni, Zhiyong Zhou, Xuemei Ge

**Affiliations:** 1Department of Food Science and Technology, College of Light Industry Science and Engineering, Nanjing Forestry University, Nanjing 210037, China; 13813972326@163.com (L.C.); l17603626554@163.com (Z.L.); luo09509816@163.com (Y.L.); 2Department of Pharmacy, Medical College of China Three Gorges University, Yichang 443002, China; yuebowen999@163.com (B.Y.); m18202658354@163.com (L.N.)

**Keywords:** curcumin, lipid, cochleates, characterization, stability

## Abstract

Curcumin (CUR), a polyphenolic substance extracted from plants, has extensive pharmacological activities. However, CUR is difficult to be absorbed in the body due to its poor stability and low solubility. Studies have found that cochleates can be used as a new delivery system to encapsulate bioactive agents for the purpose of improving its stability and bioavailability. In this study, thin-film dispersion and trapping methods were used to prepare curcumin-loaded cochleates (CUR-Cochs). Then CUR-Cochs were characterized and the encapsulation efficiency was determined by HPLC. In addition, the freeze-drying process of CUR-Cochs was studied and related characterization was performed. CCK-8 assay was used to detect the cytotoxicity of cochleates carrier. Additionally, H_2_O_2_-induced cellular oxidative damage model were used to evaluate its antioxidant capacity. The results showed that the structure of CUR-Cochs was a spiral cylinder with an average particle size of 463.8 nm and zeta potential of −15.47 mV. The encapsulation efficiency was the highest (83.66 ± 0.8)% with 1:50 CUR-to-lipid mass ratio. In vitro results showed that cochleates had negligible cytotoxicity and owned antioxidant capacity, which provided the possibility for their applications in food and medicine. In general, the method herein might be a promising method to encapsulate CUR for further use as a bioactive agent in functional foods.

## 1. Introduction

CUR, an orange-yellow crystalline powder, is a polyphenolic substance extracted from curcuma plants [1,2]. It is a natural active substance that is insoluble in water and easily soluble in organic solvents such as ethanol, methanol, and glacial acetic acid [3]. CUR is the main active ingredient in turmeric due to its multiple active groups such as phenolic hydroxyl and carbonyl [4,5]. Studies have found that CUR has numerous pharmacological activities [5,6,7]—including antioxidant, anti-tumor, promoting wound healing, and heart protection—which make it have broad prospects in food and medicine. In addition, CUR has a good clinical safety profile, and clinical trials found that CUR was well tolerated even when administered orally at a dose of 12 g per day for 3 months [8]. However, shortcomings of CUR—such as poor water solubility, poor chemical stability, high metabolic rate, and low bioavailability—have become obstacles for its further applications [9,10,11]. Among them, the bioavailability study of CUR is very important. The bioavailability of CUR is poor due to their low solubility in water and stability issues [12]. In addition, due to the hydrophobicity of CUR, only a small amount of oral CUR is absorbed by the small intestine and reaches the systemic circulation [13,14]. Furthermore, the absorbed CUR is rapidly metabolized by the liver in the body, and then excreted through the renal system, further deteriorating its physiological activity. Therefore, it is necessary to study the bioaccessibility of CUR-Cochs, mainly to investigate whether it could improve the bioaccessibility of CUR [15,16].

It is reported that many delivery systems could be used to encapsulate CUR, such as micelles, emulsion, liposome, and so on [17,18], lipid is one of the most popular active substance delivery vehicles [18]. Liposome is a spherical structure composed of a phospholipid bilayer surrounding the internal water environment, which can carry both hydrophilic and hydrophobic substances [19]. However, its mechanical stability is insufficient under long time storage and extreme environments, which makes the embedded CUR leak easily [20]. To overcome such limitations, cochleate, a new type of drug-carrying system, has attracted widespread attention of scientific researchers [21]. Cochleate is a phospholipid-cation co-precipitate, which is a cigar-liked structure, composed of the double layer of liposomes and the cationic bridging agent (Ca^2+^, Zn^2+^) by electrostatic action. Negatively charged phospholipids are often used [22]. In general, intermediate structures of cochleates during the formation process can be divided into primary, secondary, and tertiary structures. Compared with liposomes, the internal structure of cochleate is usually non-aqueous, and this multi-layered roll-like structure makes it have better mechanical stability [23,24] (Figure 1).

In this study, CUR-Cochs were prepared, and their freeze-drying process was optimized to solve the solubility and stability of CUR to meet application requirements. The preparation process of the CUR-Cochs was shown in Figure 1. We also characterized CUR-Cochs and evaluated their stability, bioaccessbility, and antioxidant properties. In this work, the stability of CUR-Cochs was compared with that of CUR-Lipos in different pH or temperature conditions. In addition, the bioaccessibility of CUR-Cochs was determined by an in vitro digestion model including mouth, stomach, and small intestine phases. The CCK-8 assay was used to investigate the cytotoxicity of Blank-Cochs. The ROS level of CUR-Cochs was also determined in this work, which was used to evaluate the antioxidant capacity. In summary, the above experimental studies on CUR-Cochs provide a basis for the further application of CUR-Cochs in food and medicine fields.

## 2. Materials and Methods

### 2.1. Chemicals and Materials

PhosphatidylSerine (PS) was purchased from Chemi Nutra Inc. (Austin, TX, USA). Phosphatidylcholine (PC) was purchased from Shanghai Macklin Biochemical Co., Ltd. Curcumin, citric acid, α-D-Lactose monohydrate, trehalose, and mannitol were purchased from Shanghai yuanye Bio-Technology Co., Ltd. (Shanghai, China). Sodium dihydrogen phosphate and dibasic sodium phosphate were purchased from Sinopharm Chemical Reagent Co., Ltd. (Shanghai, China). Artificial saliva (CZ0243, pH6.8), simulated gastric fluid (CZ0212), and artificial intestinal juice (CZ0200) were purchased from Beijing Leagene Biotechnology Co., Ltd. (Beijing, China). Cell Counting Kit-8 (CCK-8), Dulbecco’s modified Eagle medium (DMEM) and fetal bovine serum (FBS) were obtained from Dalian Meilun Biotechnology Co., Ltd. (Dalian, China). Hydrogen peroxide (H_2_O_2_) was purchased from Sinopharm Chemical Reagent Co., Ltd. (Shanghai, China). Reactive oxygen species (ROS) assay kit was purchased from Beijing Applygen Technology Co., Ltd. (Beijing, China). Ultra-pure water was obtained by MILLI-Q* ultra-pure water system.

### 2.2. Preparation of CUR-Cochs by Trapping Method

In this work, thin-film hydration method was used to prepare CUR-Lipos. PS and CUR were dissolved in dichloromethane in a 250 mL eggplant-shaped flask, which was connected to SY-2000* rotary evaporator (Shuangxu Electronics Co., Ltd., Shanghai, China) to evaporate the solvents at 25 °C with a speed of 60 rpm for 20 min to form a uniform film. Then, the flask was placed in a vacuum drying dish for more than 2 h and blew nitrogen for 5 min to remove residual organic solvents. The film was hydrated with 45 mL ultra-pure water by magnetic stirrer for 1 h above the T_m_ of PS followed by sonicating for 30 min, the impurities were removed by Sigma 2-16k centrifuge (BMH Instrument Co., Ltd., Beijing, China) with 6000 rpm and at 4 °C for 10 min. Finally, transparent and uniform dispersion of CUR-Lipos were obtained by extrusion through a polycarbonate membrane with 0.45 µm pore size. CUR-Cochs were formed by trapping method, which was formed by adding CaCl_2_ solution (50 mM) slowly (10 µL at a time) to CUR-Lipos with constant low-speed stirring above T_m_ of the lipid.

### 2.3. Study on the Freeze-Drying Process of CUR-Cochs

The freeze-drying technology was used to prepare freeze-dried CUR-Cochs to increase its storage stability. 4 mL samples were placed in a 10 mL vial respectively, and then different freeze-drying protective agents were added to ensure that the physical and chemical properties of samples remained unchanged before and after freeze-drying. Lactose, mannitol, trehalose, and glucose (setting three concentrations of 5%, 10%, and 15% (*w*/*v*) respectively) were selected as freeze-dried protective agents. Firstly, prepared sam ples were put in a −80 °C refrigerator to pre-freeze for 24 h. After that, samples were put into a Scient-18N* vacuum freeze dryer (Sciebtz Biological Technology Co., Ltd., Ningbo, China) for 24 h at −55 °C and a vacuum pressure under 100 Pa to obtain CUR-Cochs freeze-dried powder.

### 2.4. Encapsulation Efficiency Analysis

CUR-Cochs with different CUR-to-lipid mass ratios (1:30, 1:40, 1:50, 1:60) were prepared, and the encapsulation efficiency (EE) was determined to optimize the best condition. The HPLC was used to determine the content of CUR. The HPLC condition could be set up as below: HPLC Column: Welchrom^®^ C18 (5 µm, 4.6 × 150 mm); Mobile phase: methanol:1% citric acid (70:30, *v*:*v*); Column temperature: 30 °C; Flow rate: 1 mL/min; Detector: UV detector; Detection wavelength: 424 nm (CUR); Injection volume: 10 µL.

The determination of total CUR in CUR-Cochs was carried out as follows: Firstly, 200 μL dichloromethane was added in 0.5 mL CUR-Cochs solution and the mixed solution was sonicated for 5 min. Then the demulsified solution was adjusted to 5 mL with methanol followed by extrusion through the 0.22 μm organic microporous membrane and placed in Waters e2695 HPLC (Waters Technology Co., Ltd., Shanghai, China) to calculate the total CUR content. Determination of CUR encapsulated in CUR-Cochs: 0.5 mL CUR-Cochs solution was centrifuged at 6000 rpm for 5 min to obtain the precipitate, which was mixed with 200 μL dichloromethane to ultrasonic for 5 min. Then the following steps were the same as above and the final results were obtained by HPLC. EE was calculated as follows in Equation (1). W_e_ is the concentration of CUR encapsulated in CUR-Cochs, and W_f_ is the concentration of total CUR.
EE (%) = W_e_/W_f_ × 100(1)

### 2.5. Characterization of CUR-Cochs

#### 2.5.1. Particle Size and Zeta Potential Analysis

Firstly, 1 mL CUR-Lipos was added to a 2 mL centrifuge tube respectively, and then different volumes of 50 mM CaCl_2_ solution was taken in it under the condition of magnetic stirring and above T_m_. Additionally, the average particle size and the zeta potential of different samples were detected by the Malvern Nano ZetaSizer ZS (Malvern Instruments, Ltd., Worcestershire, UK). The experimental temperature was set to 25 °C. Then the equilibrium time of samples was set to 120 s and the data obtained each time was the average value of 15 times by the instrument. Each sample was measured three times.

#### 2.5.2. TEM Observation

TEM can be used to observe the morphology of samples. Firstly, a droplet of sample was placed on a carbon-coated copper grid for 4 min. The copper grid was put on a lint-free filter paper to absorb the excess liquid and air dried overnight. At last, the dry copper grid was observed using a JEM-1400 TEM.

#### 2.5.3. DSC Analysis

DSC was used to determine the thermal behavior of the CUR-Cochs. Raw materials of PS, CUR and mannitol and CUR-Cochs freeze-dried powder were respectively weighed in a certain amount (5–15 mg) and placed in an aluminum crucible with a thin layer on the bottom. The aluminum lid was pressed tightly and pierced with a small hole. In addition, the crucible containing the sample was placed inside placed in the calorimeter and compared with the empty one. In case of the heating rate of 10 °C/min, DSC scan results were performed. Finally, the thermal properties of samples were analyzed according to their heat flow curves.

### 2.6. Study on the Stability of CUR-Cochs

#### 2.6.1. Stability of CUR-Cochs against pH

Phosphate buffered saline (PBS) solutions with different pH values (5.5, 6.5, 7.4, 8, and 9) were prepared by sodium dihydrogen phosphate and dibasic sodium phosphate. Then CUR-Lipos and CUR-Cochs were added to PBS solutions respectively, and incubated in a dark water bath at 37 °C. After 2 h, samples were dissolved with methanol to detect the content of CUR by HPLC to evaluate whether cochleates could improve the stability of CUR. The CUR content of original solution was taken as 100% to calculate the CUR content changes at different pH.

#### 2.6.2. Stability of CUR-Cochs in Different Temperature

The effect of heating for different temperatures on the stability of CUR-Cochs was examined comparing with the CUR-Lipos group. 4 mL CUR-Lipos and CUR-Cochs solutions were added into 5 mL vials in five different groups, and then heated in a water bath at 37, 40, 60, 80, and 100 °C respectively for 2 h. Among them, the freshly prepared CUR-Cochs solution stored at 4 °C was used as the control group. After the treatment, methanol was added to demulsify in each vial, and then the content of CUR was measured by HPLC. The CUR retention rate was determined to investigate the thermal stability of CUR-Cochs. The CUR content of original solution was taken as 100% to calculate the CUR content changes at different temperature.

### 2.7. In Vitro Bioaccessibility of CUR-Cochs

#### 2.7.1. Simulated Gastrointestinal Digestion

In this study, the bioaccessibility of CUR in CUR-Cochs was determined by an in vitro digestion model, which was a simulated gastrointestinal tract consisting of mouth, stomach, and small intestine phase [25,26].

Mouth phase—The prepared CUR-Cochs suspension was poured into a conical flask, then was preheated in a constant temperature water bath at 37 °C. The prepared artificial saliva fluid was poured into the conical flask and mixed with CUR-Cochs at a mass ratio of 1:1, and the pH of the mixed solution was 6.8. Subsequently, the resulting mixed solution was placed in a shaker and shaken with the speed of 90 rpm for 10 min at 37 °C to simulate oral conditions.

Stomach phase—Firstly, simulated gastric juice containing pepsin was prepared and the pH value was 2.5. Then the simulated gastric juice was added to the sample produced by the mouth phase at a mass ratio of 1:1, and the mixture continued to be cultured in a shaker for 2 h at a speed of 100 rpm in 37 °C.

Small intestine phase—Samples in the simulated gastric phase were adjusted to pH 7.0 with 1 M NaOH solution, and then the prepared small intestinal fluid was added. The pH of the resulting mixture was kept constant at pH 7.0 by adding 50 mM NaOH solution. The samples were kept for 2 h at 37 °C.

#### 2.7.2. CUR Stability and Bioaccessibility after Digestion

After digestion, CUR dissolved in the mixed micellar phase was generally considered to be available for assimilation, since certain mixed micelles were small enough to be transported across the mucus layer to epithelial cells [27]. Therefore, the determination of CUR content in the mixed micelle phase was the bioaccessibility of CUR. After the digestion of the small intestine, an appropriate amount of the digestive juice was allowed to centrifuge with 15,000 rpm for 30 min at 4 °C, the obtained supernatant was a mixed micelle phase, and the CUR content was measured according to the method in 2.4. The stability and bioaccessibility of CUR in CUR-Cochs were calculated by the equations
Stability (%) = C_Digesta_/C_Initial_ × 100(2)
Bioaccessibility (%) = C_Micelles_/C_Digesta_ × 100(3)

Herein, C_Initial_ is the initial concentration of CUR in cochleates before digestion reaction, C_Digesta_ is the concentration of CUR in overall raw digesta, C_Micelles_ is the concentration of CUR dissolved in micelles fraction.

### 2.8. Cytotoxicity of Blank-Cochs by CCK8 Assay

Cell Counting Kit-8 (CCK-8) assay was performed to detect cytotoxicity. Firstly, cells were added into 96-well plates at the density of 1 × 10^4^ cells/mL (200 µL/well). After incubated at 37 °C in 5% CO_2_ incubator for 24 h, 0.4, 0.8, 1.2, 1.6, and 2 M Blank-Lipos and Blank-Cochs solutions were added to each well, and then incubated for 24 h. Before measuring the cytotoxicity, the 96-well plate was placed under an inverted microscope to observe the cell morphology. The culture medium was replaced with 10% CCK-8 solution in 200 μL medium and co-incubated with cells for another 2 h. The absorbance at 450 nm was recorded using a microplate reader, which represented the cell viability. Cell viability (%) was calculated by using Equation (4). Among them, As represented the absorbance of experimental wells, Ac represented absorbance of control wells and Ab represented the absorbance of blank wells.
Cell viability (%) = (As − Ab)/(Ac − Ab) × 100(4)

### 2.9. Intracellular ROS Assay of CUR-Cochs

The antioxidant injured model was set up to determine the protection effects of the CUR-Cochs as described [28]. First of all, NIH3T3 cells were seeded in a 6-well plate and cultured in an incubator with 5% CO_2_ at 37 °C for 24 h. Then the culture medium was discarded, washed with PBS 2 or 3 times, and 2 mL culture medium containing the sample was added separately. The normal control group (only the medium without H_2_O_2_ solution and samples), model group (the medium without samples directly treated with H_2_O_2_ solution) and test groups were set up. Among them, test groups were also added 2 mL culture medium including CUR-Cochs solutions which contained 0.75, 1.5 and 3 µM CUR, 3 µM CUR solution (dissolved in DMSO), and Blank-Cochs solution. After samples were processed for 12 h, 600 µM H_2_O_2_ solution was added to each well for 8 h. Then the cells were washed with PBS and collected in centrifuge tubes, and then 1 mL culture medium containing 10 µM 2′,7′-dichlorodihydrofluorescein diacetate (DCFH-DA) was added respectively. After incubating in the incubator for 30 min in the dark, the residual probes were washed with PBS, cells were collected and resuspended in PBS, and then the ROS value was measured by flow cytometry (Becton, Dickinson and Company, Franklin Lakes, NJ, USA).

### 2.10. Statistical Analysis

All experiments were performed in triplicate and results are expressed as mean ± SD. The data was analyzed with SAS 9.2 software. The single factor method was used to analyze the significance of the difference, and the least significant difference was compared in pairs. A *p*-value of <0.05 was considered statistically significant. All pictures were made with Origin 8.5 and AI 2018 software.

## 3. Results and Discussion

### 3.1. Preparation Process of CUR-Cochs

CUR is a natural polyphenol compound, which can be used as food additives such as colorants and antioxidant functional component. However, low solubility, poor stability and low bioavailability have been the bottleneck for further development and utilization of CUR. The preparation of CUR-loaded delivery systems from edible natural sources—such as proteins, polysaccharides, and lipids—is an effective strategy to solve the above problems. The cochleate, as an emerging delivery technology, with better stability and bioavailability, is the optimal choice for curcumin delivery. In this work, the trapping method was used to prepare CUR-Cochs. CaCl_2_, as a cationic linking agent, can bind to the negatively charged head group of CUR-Lipos to open the original structure of the liposome and form a new cochleate structure. Therefore, the content of CaCl_2_ is critical to the formation of cochleates (Figure 1). Firstly, 10–100 µL CaCl_2_ was added to the CUR-Lipos solution respectively with constant low-speed stirring at 60 °C to observe the formation process of cochleates. After 1 h of reaction, the CUR-Cochs precipitate was collected by low-speed centrifugation, and then resuspended in ultrapure water to form a suspension, then were characterized below. When 80 µL CaCl_2_ (50 mM) was dropped into CUR-Lipos solution, the structure of CUR-Cochs could be found through an optical microscope (Appendix A). Additionally, much more effort should focus on the feasibility of the scale up as well as the standards of reproduction from batch to batch.

### 3.2. Characterization of CUR-Cochs

#### 3.2.1. Particle Size and Zeta Potential Analysis

The particle size and zeta potential of samples were determined by the Malvern Nano ZetaSizer ZS. Studies found that with the volume of CaCl_2_ increased, the zeta potential of CUR-Lipos solution gradually decreased and the particle size of CUR-Cochs continued to increase. This was because the addition of CaCl_2_ made the bilayer membranes of CUR-Lipos fuse with each other to form aggregates, and then Ca^2+^ and the negatively charged lipid head groups combined with each other through electrostatic interaction and was curled into CUR-Cochs with a cochlear structure. When the volume of CaCl_2_ was more than 40 µL, the zeta potential of CUR-Cochs was around −15 mV (Figure 2A). The particle size results showed that when the volume of CaCl_2_ was 100 µL, the average particle size of CUR-Cochs was 540 nm. Finally, combining results and the need of the experiment, 90 μL of CaCl_2_ was selected to prepare CUR-Cochs with 463.8 nm and −15.47 mV.

#### 3.2.2. TEM Observation

As shown in Figure 2B, we could find the provided details on morphology of cochleates by observation of TEM. The cochleate was a cigar-like shape and the particle size was around 300 nm, which was comparable with the size distribution results obtained from Malvern Nano ZetaSizer ZS analysis. Moreover, the structure of cochleate was long columnar, and from the two ends of the cochleate, we could see that the internal structure of it was multi-layered.

### 3.3. Screening of CUR-to-Lipid Ratio

In this experiment, ultracentrifugation was used to separate cochleates and free CUR. In addition, HPLC was used to determine the content of CUR, and finally the encapsulation efficiency was obtained. Firstly, the chromatogram of the reference solution was shown in Appendix A, and we could find that the retention time of CUR was 6.7 min. The standard curve of the CUR content measured by HPLC was y = 87589x − 13741, R^2^ = 0.9998 (2.5~50 μg/mL) (Figure 2C). As shown in Figure 2D, when the CUR-to-lipid ratio was 1:50, the encapsulation efficiency of CUR-Cochs could reach the highest, which was 83.66 ± 0.8%.

### 3.4. Selection of Freeze-Dried Protective Agent of CUR-Cochs

#### 3.4.1. Appearance Observation

Lactose, mannitol, trehalose, and glucose (with the mass to volume ratio of 5%, 10%, 15%) were selected as freeze-drying protective agents. It was found that the freeze-dried effect of the mannitol group was the best which observed from the appearance of freeze-dried samples, and the results were shown in Table 1. The freeze-dried samples of the lactose and trehalose groups both collapsed and slightly shrank in appearance, and the powder was rough and uneven with large particles. The glucose group was severely atrophied, and the mannitol group had the best freeze-drying protecting effect which had a smooth and full appearance, and the powder was fine and evenly dispersed (Appendix A).

#### 3.4.2. Particle Size and Zeta Potential

From Figure 3A, we could find that CUR-Cochs powder containing 5% mannitol had the best solubility in water with uniform particle size dispersion. After re-dissolved in water, the average particle size was 422.9 nm, and the zeta potential was −23.3 mV, which was similar to the results of CUR-Cochs original solution, and there was no significant difference (*p* < 0.05). However, the particle size distributions of samples with 10% mannitol or 15% mannitol were not uniform, and there was significant difference in zeta potential between 15% mannitol CUR-Cochs and CUR-Cochs original solution. According to the experiment, 5% mannitol was selected as the freeze-drying protective agent. The difference in zeta potential values after addition of mannitol was caused by the physical characteristic changes of the CUR-Coch. With adding 5%, 10%, and 15% concentration of the mannitol as freeze-drying protect agents, the ability of re-dispersion after dissolve in water of the CUR-Coch powder was different. The optimized 5% mannitol group shown the no significant difference of in size and zeta potential (−23.3 mV) with the original one (−20.8 mV). For the 10% and 15% mannitol groups, the particle sizes were increased to micro sizes and the aggregation can be observed. The forming of larger particle sizes as well as the aggregation may lead to the shielding of more negatively surface charges and probably the reason to cause the gradually reduced zeta potential (−17.2 mV in 10% group and −14.3 mV in 15% group).

#### 3.4.3. SEM Observation and DSC Analysis

As shown in Figure 3B, the morphology of CUR-Cochs freeze-dried powder was cigar-liked long columnar structures by SEM observation. In addition, DSC could be used to analyze the composition of the sample by different substances having unique melting points at a certain temperature. In this work, DSC was used to analyze the thermal properties of carrier materials, freeze drying protectant, CUR, and CUR-Cochs, and the results were shown in Figure 3C. The results shown that the melting endothermic peak of phosphatidylserine was at 71.49 °C, the endothermic peak of CUR was at 187.50 °C and the endothermic peak of mannitol was at 171.77 °C. After forming CUR-Cochs, the endothermic peak moved forward to 170.75 °C, indicating that the carrier formed a new phase, rather than a simple physical mixture. At the same time, the melting peak of CUR (187.50 °C) disappeared completely, indicating that CUR was encapsulated in CUR-Cochs with a non-free state.

### 3.5. Study on the Stability of CUR-Cochs

The chemical properties of CUR itself are unstable due to many pairs of carbon-carbon double bond structures in its molecular structure. Therefore, CUR is sensitive to environmental conditions such as high temperature, light, extreme pH, and so on. In this work, PS was used as a material to prepare cochleates as a bio-delivery system to embed CUR and improve the stability. This work discussed that whether cochleates encapsulation could improve the stability of CUR comparing with liposome.

#### 3.5.1. Stability against pH of CUR-Cochs

It has been demonstrated by many studies that the decomposition of CUR depended on different pH values, and CUR was relatively stable under acidic to neutral conditions but degraded faster in alkaline environment [29]. Under weakly acidic and neutral conditions, the ketone form of CUR was dominant. However, in the term of alkaline condition, CUR mainly existed in the form of enol, which has the ability to scavenge free radicals [30]. In this work, CUR residue rates of CUR-Lipos and CUR-Cochs at different pH values (5.5, 6.5, 7.4, 8, 9) were determined, and CUR-Lipos or CUR-Cochs prepared with ultrapure water was used as a control group respectively. As shown in Figure 4A, the retention rates of CUR in CUR-Cochs were significantly higher than CUR-Lipos with the same pH values (*p* < 0.05). In addition, it could find that the stability of samples was low at pH 9. The results showed that the pH stability of CUR-Cochs was significantly improved compared to CUR-Lipos (*p* < 0.05). Under neutral or weakly acidic conditions, samples had better stability, while CUR in samples were extremely unstable under alkaline conditions.

#### 3.5.2. Thermal Stability of CUR-Cochs

CUR is a heat-sensitive biologically active substance, so it was essential to test the thermal stability of CUR-Cochs. Cochleates solutions may face different temperatures during processing, transportation, and storage during the application of the food industry. In this study, CUR retention was used to reflect the thermal stability of the samples in different temperature. As shown in Figure 4B, the retention rate of CUR on 37 °C and 50 °C groups were almost close to 100% compared with control group, which meant that CUR-Lipos and CUR-Cochs were stable in human body temperature and also stable in 50 °C so that it would not be destroyed. In addition, as the temperature gradually increased (60–100 °C), the retention rate of CUR in CUR-Lipos and CUR-Cochs showed a decreasing trend, and when the temperature reached 100 °C, the retention rate of CUR-Cochs reached 67.5%. In addition, we could find that the retention rate of CUR in CUR-Cochs was significantly higher than that in CUR-Lipos (52.5%) at 100 °C (*p* < 0.05). This showed that high temperature still had a certain effect on the stability of CUR-Cochs, but was more stable than CUR-Lipos. In general, the structure of cochleates was stable, which could well protect the CUR in the phospholipid bilayer and significantly improved the thermal stability of free CUR [31]. At high temperature, the stability of CUR-Cochs was better than that of CUR-Lipos. Comparing with liposome, the mechanical stability of the cochleats could be improved, which may offer one strategy to serve as a lipid-based bioactives delivery system to meet the requirement of stability with low cytotoxicity.

### 3.6. In Vitro Bioaccessbility of CUR-Cochs

Bioavailability refers to the speed and extent to which the active ingredient enters the human circulation after being absorbed. CUR has many biological and pharmacological activities that may benefit human health [32]. However, CUR has a very low bioavailability due to its poor oral absorption, rapid metabolism, and elimination in the body [33,34]. Traditional oral administration could not exert the curative effect of CUR well, so researchers have explored a variety of methods to improve the bioavailability of CUR in recent years [35,36,37]. In this study, the in vitro bioaccessbility of free CUR and the CUR encapsulated in cochleates was compared using a simulated gastrointestinal tract (GIT), and the results were expressed in terms of stability and bioaccessibility. As shown in Figure 4C, the content of free CUR (72.17%) in the overall digesta after exposure to the simulated GIT was higher than that of CUR-Cochs (61.92%), which indicated that free CUR was more stable than CUR-Cochs in the gastrointestinal digestive juice. The free CUR used in this study was dissolved in methanol and then suspended with ultrapure water forming relatively large CUR crystals. The free CUR was expected to have limited specific surface area compared with CUR-Cochs. Therefore, CUR loaded in cochleates was more easily exposed to the surrounding aqueous phase, leading to more chemical degradation. In addition, the bioaccessbility of encapsulated CUR (58.77%) in cochleates was significantly higher than that of free CUR (25.58%), indicating that CUR encapsulated in cochleates was more easily dissolved in mixed micelles than free CUR. This may be due to the amorphous of CUR in CUR-Cochs and better water solubility of CUR-Cochs, which could accelerate their dissolution in micelles. In conclusion, the in vitro bioaccessbility of the CUR could be greatly enhanced by encapsulating it into cochleates.

### 3.7. Measurement of Cytotoxicity of Blank-Cochs

The cytotoxicity of Blank-Lipos and Blank-Cochs to L929 cells was detected by the CCK-8 assay. As shown in Figure 5A, it was found that the cell morphology of the experimental groups grew well, which was indistinguishable from the control group. In addition, the results of the cytotoxicity experiments shown in Figure 5B indicated that the cell viability of L929 cells treated by Blank-Lipos and Blank-Cochs were higher than 100% at all concentrations (0.4~2 M), which suggested that carriers of liposomes and cochleates prepared in this work both had negligible toxicity.

### 3.8. Intracellular ROS Assay of CUR-Cochs

ROS plays a dual role in the biological system. Low levels of ROS can participate in the regulation of cell survival signals, and high levels of ROS can damage cell structures and cause oxidative damage or even apoptosis, which is often the main cause of cellular oxidative damage [38,39]. In this study, the DCFH-DA fluorescent probe method was used to detect the effects of CUR and CUR-Cochs on the level of intracellular ROS. As shown in Figure 5C, ROS production was less in control group. When cells were stimulated with H_2_O_2_ injury, the level of ROS increased to approximately 390% of that in control group. Comparing with the H_2_O_2_ group, there was no obvious change in the Blank-Cochs group. Both CUR and CUR-Cochs groups could reduce the increase of cellular ROS induced by H_2_O_2_, and the CUR-Cochs treatment group showed a good dose-dependence. At the concentration of CUR was 3 μM, the ROS level of the CUR and CUR-Cochs groups were 131% and 150% of that in control group respectively. The above experimental results showed that both CUR and CUR-Cochs had antioxidant effects, thereby protecting cells from oxidative damage. Among them, the protective effect of CUR was slightly better than that of CUR-Cochs, because CUR in this work was dissolved by DMSO and had better dispersibility. In fact, CUR is insoluble in water without adding DMSO solvent when applied. Future research could mostly focus on how to evaluate the potential toxicity of the CUR-Coch nanoparticle as in vivo delivery system. Additionally, considerable effort should be involved to investigate the mechanism of the CUR-Coch for their further application.

## 4. Conclusions

In this work, PS was used as the carrier material, and the film dispersion method together with the ultrasonic hydration method was used to prepare CUR-Lipos, and then the trapping method was used to formulate CUR-Cochs. The encapsulate rate of CUR in cochleates was about 83.66% with 1:50 CUR-to-lipid mass ratio. The prepared CUR-Cochs were characterized with an average particle size of 463.8 nm and zeta potential of −15.47 mV. In addition, the freeze-drying process of CUR-Cochs was also studied. The results showed that the freeze-dried powder formed with 5% mannitol as the freeze-dried protective agent had the best effect. Comparing with CUR-Lipos, the results revealed that the stability of CUR loaded in CUR-Cochs could significantly improve in different pH and temperature (*p* < 0.05). In addition, the results indicated that CUR-Cochs could improve the bioaccessbility of CUR. Moreover, the cytotoxicity experiments showed that PS as the carrier material was non-toxic by CCK-8 assay. The study of intracellular ROS assay has shown that CUR-Cochs could maintain the ability of anti-oxidation of the CUR and reduce the ROS level in the oxidative damage model with a dose-dependent effect. In general, the prepared CUR-Cochs exhibit the ability of antioxidant with relative low cytotoxicity, which could be potentially to be developed as functional foods or as foods additives. The mechanism of the CUR-Cochs for their antioxidant effect should be further investigated for better application. Additionally, more experiments should be performed to optimize the suitable condition from batch to scale up for their applications. Cochleates technology could also provide a new idea and method for improving the performance of hydrophobic biologically active substances.

## Figures and Tables

**Figure 1 foods-11-00710-f001:**
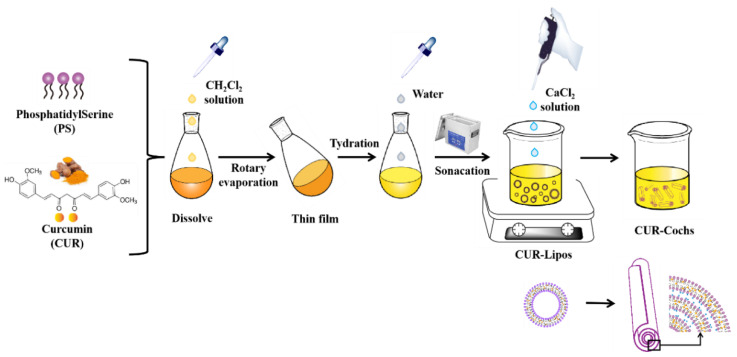
Preparation process of CUR-Lipos and CUR-Cochs.

**Figure 2 foods-11-00710-f002:**
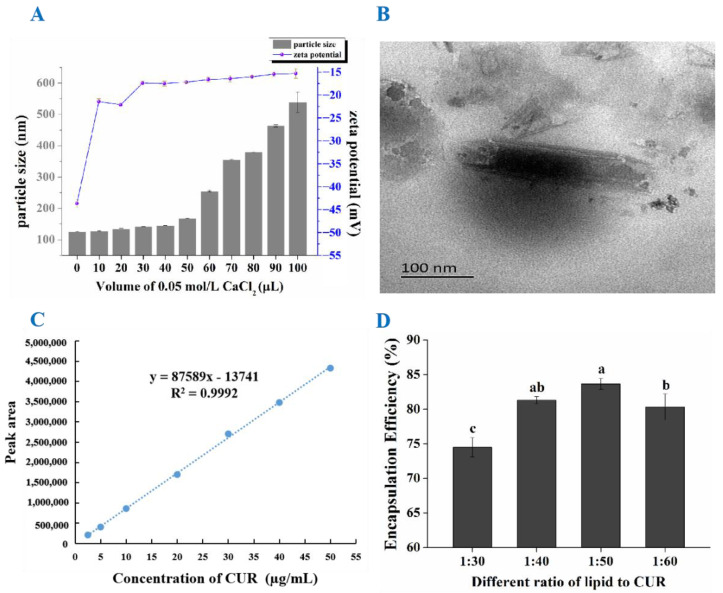
(**A**) Particle size and zeta potential of samples with different volumes of CaCl_2_. (**B**) TEM observation of CUR-Cochs. (**C**) The standard curve of CUR. (**D**) Encapsulation efficiency of CUR-Cochs with different CUR-to-lipid ratio. Different lowercase letters indicate significant differences (*p* < 0.05).

**Figure 3 foods-11-00710-f003:**
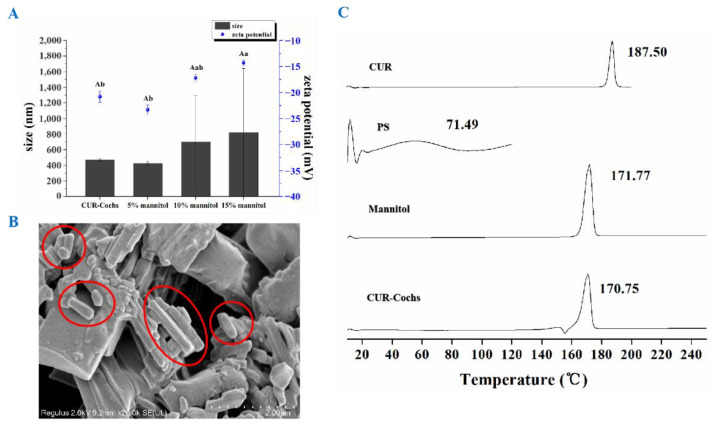
(**A**) Particle size and zeta potential distribution of CUR-Cochs. Different capital letters indicate significant differences (*p* < 0.05) of particle size. Different lowercase letters mean significant differences (*p* < 0.05) of zeta potential. (**B**) Image of CUR-Cochs freeze-dried powder with SEM. (**C**) The DSC diffraction patterns of different samples.

**Figure 4 foods-11-00710-f004:**
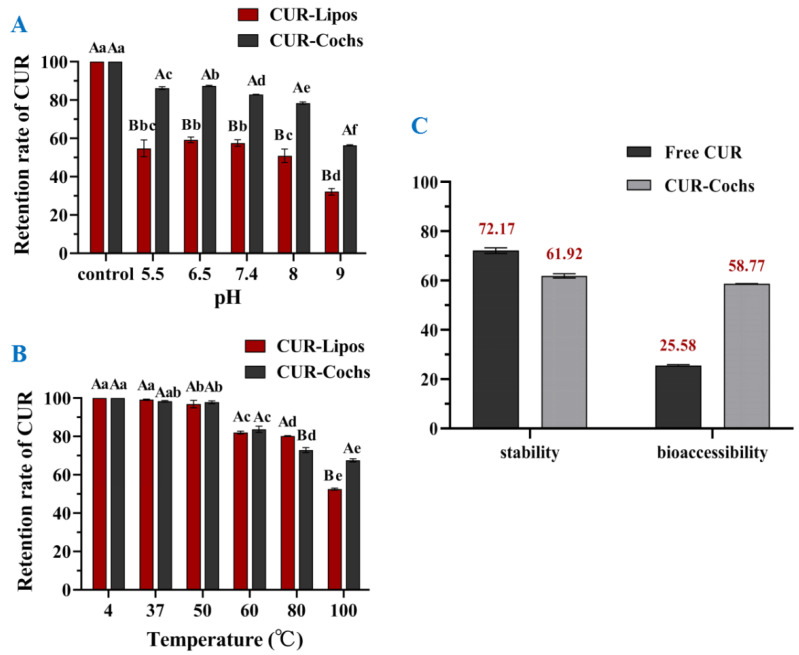
Stability of CUR-Lipos and CUR-Cochs in conditions of different (**A**) pH and (**B**) temperature. Different lowercase letters indicate significant differences (*p* < 0.05) of retention rate of CUR at different pH or temperature; Different capital letters mean significant differences (*p* < 0.05) of retention rate of CUR between CUR-Lipos and CUR-Cochs at the same condition. (**C**) The stability and bioaccessibility of free CUR and CUR-Cochs after going through a simulated gastrointestinal reaction.

**Figure 5 foods-11-00710-f005:**
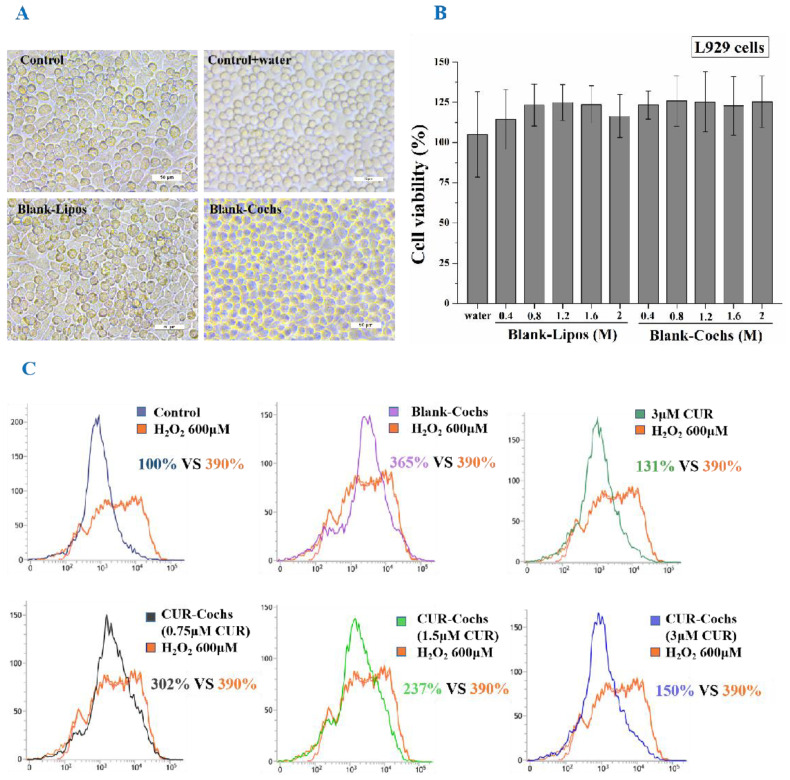
Cytotoxicity of blank carriers to L929 cells by CCK-8 assay and antioxidant effects. (**A**) Schematic diagram of cell morphology. (**B**) L929 cells viability of control group, Blank-Lipos, and Blank-Cochs groups. (**C**) ROS levels in NIH3T3 cells were evaluated by flow cytometry.

**Table 1 foods-11-00710-t001:** Effect of different protective agents.

Protective Agents	Dosage (*w*/*v*)	Appearance
Lactose-1	5%	Slightly shrunken, yellow
Lactose-2	10%	Slightly shrunken, yellow
Lactose-3	15%	Slightly shrunken, yellow
Mannitol-1	5%	Smooth and full, light yellow
Mannitol-2	10%	Smooth and full, light yellow
Mannitol-3	15%	Smooth and full, light yellow
Trehalose-1	5%	Slightly collapsed, yellow
Trehalose-2	10%	Slightly collapsed, yellow
Trehalose-3	15%	Slightly collapsed, yellow
Glucose-1	5%	Severely shrunken and adherent, yellow
Glucose-2	10%	Severely shrunken and adherent, yellow
Glucose-3	15%	Severely shrunken and adherent, yellow

## Data Availability

Not applicable.

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
