# Peer review of "Study on the Preparation, Characterization, and Stability of Freeze-Dried Curcumin-Loaded Cochleates"

_foods, 2022, doi:10.3390/foods11050710_

Round 1

Reviewer 1 Report

In this work, the authors explore CUR-chochs as delivery systems of curcumin. The work is very interesting, however same important methods are not reported:

  • Low speed conditions used to remove impurities (lines 101-102);
  • Ratios between curcumin and lipids (line 119);
  • HPLC conditions, e.g., mobile phase, flow rate and UV detector,

Regarding the results, some of them are not very well justified:

  • Differences found in zeta potential values after addition of mannitol (section 3.4.2)

Author Response

请参阅附件。

Reviewer 2 Report

The paper presented for the review proposes a new method to prepared curcumin-loaded cochleates ,  evaluates stability, bioaccesibility of obtained cochleates and characterizes  their cititoxity and antioxidant cacacity. The scientific contribution of this paper is at a satisfactory level and the approach and methodology are appropriate for the purpose of the investigation, but some suggestion and comments should be considered by the authors:

In the introduction section the authors in the last paragraph sholud only state the aim, purpose and methodology of the research. Line 62-64, and Line 72-76 The authors present results and findings of the work. This sentences should be presented in the section for discussion or conclusions.  Please rearrange this sentences.

Line 64-71 Move this section explaining  bioavailability of CUR to the firts paragraph of the introduction.

Line 94-107 Please, scpecify the producer, city and country for rotary evaporator and centrifuge.

Line 370-377 Add references.

At the end of the discussion, write in which direction further research should be done and indicate the potential for application in food.

Reviewer 3 Report

In the presented manuscript, curcumin-loaded cochleates were prepared by thin-film dispersion methods, characterized, freeze-dried, and optimized, stability, encapsulation efficiency, antioxidant capacity, and in vitro bioaccessibility were determined. The research is a comprehensive manuscript well-written, the results are nicely presented and properly discussed, I have only a few comments.

Page 9 and page 10, discuss in more detail the thermal stability of CUR-cochs.

Page 12, In the discussion section, discuss the results from the perspective of previous studies, compare it with the relevant literature. What are the future research directions?

Page 13, It is not mandatory; however consider adding a conclusion to summarize the presented findings.

Round 2

Reviewer 1 Report

The replies are well justified.